# Special-Oriented Annealing-Twins-Induced Orange Peel Morphology of Heat Pipe under Bending Deformation

**DOI:** 10.3390/ma16062147

**Published:** 2023-03-07

**Authors:** Song-Wei Wang, Hong-Wu Song, Shi-Hong Zhang, Shuai-Feng Chen

**Affiliations:** Shi-changxu Innovation Center for Advanced Materials, Institute of Metal Research, Chinese Academy of Sciences, Shenyang 110016, China

**Keywords:** surface roughening, heat pipe, annealing twin, texture

## Abstract

The thin-wall heat pipe is an efficient heat transfer component that has been widely used in the field of heat dissipation of high-power electronic equipment in recent years. In this study, the orange peel morphology defect of thin-wall heat pipes after bending deformation was analyzed both for the macro-3D profile and for the micro-formation mechanism. The morphology and crystal orientations of the grains and annealing twins were carefully characterized utilizing optical metallography and the electron backscatter diffraction technique. The results show that after high-temperature sintering treatment, the matrix grains of the heat pipe are seriously coarsened and form a strong Goss texture, while certain annealing twins with the unique copper orientation are retained. The distribution of the Schmid factor value subjected to the uniaxial stress indicates that inhomogeneity in the intergranular deformation exists among the annealing twins and matrix grains. The annealing twin exhibits a “hard-oriented” component during the deformation; thus, it plays a role as a barrier and hinders the slipping of dislocation. As the strain accumulates, part of the annealing twins may protrude from the surface of the heat pipe, forming a large-scale fluctuation of the surface as the so-called “orange peel” morphology. The 3D profile shows the bulged twins mostly perpendicular to the drawing direction, about 200–300 in width and 10–20 μm in height.

## 1. Introduction

The heat pipe is widely used in the field of electronic appliances (such as computers, fans, and cell-phone, etc.) due to its high heat conduction efficiency. However, the surface roughening problem, the so-called “orange peel”, after bending deformation of the heat pipe, has been raised as a limitation for its application. Orange peel defects as a surface roughening phenomenon have drawn great attention due to their negative effects on the forming process and product quality, which have been generally noticed in science and industry [1]. This type of surface defect has the following disadvantages: (1) affects the beauty of the product, which makes it difficult to use in the field of high-end equipment heat dissipation; (2) decreases the heat transfer efficiency by affecting the fitting degree of the outer surface of the heat pipe and the connecting surface, which readily produces voids, thus affecting heat transfer; and (3) decreases the surface quality after bending deformation. Thus, the specification and control of an unsuitable surface topography is an essential manufacturing requirement. Therefore, figuring out the influencing factors is of great importance for research purposes and manufacturing control.

Surface roughening problems for FCC metals, i.e., aluminum alloys and steels, have been extensively studied by both experimental and numerical methods. The influence of the microstructure aspect (such as grain size and texture) and the loading condition (such as temperature and strain rate) on the orange peel, ridging, and roping has been researched. Many results have confirmed that surface roughening is strongly dependent on the grain size of materials during deformation [2,3,4,5]. Dai investigated the plastic-deformation-induced surface roughening mechanism of aluminum sheets. The average grain rotation and grain size were found to be the dominant contributor to the surface vertical characteristics, such as the root-mean-square roughness. The surface horizontal characteristics, such as the correlation length, were found to be mainly determined by the average grain size. In addition, studies on the relationship between plastic strain and deformation-induced roughness were carried out [6,7,8,9]. Cai et al. [9] presented an interesting approach to obtain continuous strain distribution on a sample after deformation. By using a specially designed stage, it was possible to measure changes in surface roughness in situ for increasing tensile load. The results showed that the surface roughness increased when the strain was below a certain value and slightly decreased as the strain increased. Recently, the investigations [10,11,12,13,14,15] have focused on understanding the formation mechanism of surface roughness. Utilizing the electron backscatter diffraction (EBSD) technique, the crystallographic orientation of grains on the surface can be characterized. In addition, this information can be directly incorporated into the finite element method (FEM) using simulation software, such as ABAQUS. Kishimoto et al. [12] found that the crystal orientation suppresses the development of the inner surface roughness of the micro-tube, which can be investigated by evaluating the crystal orientation of the inner surface when increasing and decreasing the wall thickness during the new hollow sinking process. The results showed that the outer surface roughness developed during the new hollow sinking because of the excessive thinning of the outer diameter. Wu et al. [13] developed a 3D crystal plasticity finite element model to simulate the development of a rope-like surface roughness profile, and the results demonstrated that the 3D spatial distribution of specific grain orientations was the determinant factor for surface roughening. Similar results were observed by Wilson [14] during hydro-forming and bending deformation. Romanova et al. [15] proposed a three-dimensional model of the material with a hardened layer of varying thickness and strength compared with the initial material. A numerical analysis was carried out with the finite difference method. It was found that internal stresses appearing at the layer boundary and bulk material were responsible for surface roughness, which increased as the hardened layer thinned. It was observed that the local increase in irregularities was due to the movement of connected grains.

Although there have been many studies on the surface roughening problem represented by the orange peel defect, its formation mechanism is still worth discussing due to the complexity of the deformation materials and heat treatment process. The thin-walled heat pipes discussed in this paper have a unique sintering process and unique structural characteristics. After the high-temperature heat treatment, the grains substantially grow through the recrystallization and growth stage, and especially, a large number of annealing twins are generated in pure copper. The effects of grain size and annealing twins on the surface roughening need to be investigated, as these studies are lacking in the literature. In practical applications, these heat pipes are usually bent into various shapes as required; thus, the orange peel morphology on the outer surface needs to be addressed to improve the quality of heat pipe products. It would be meaningful to better understand the formation mechanism of the surface defect and to provide a reference for production.

## 2. Materials and Methods

The material used for heat pipes was pure copper with a composition of 99.996%. The size of the heat pipes was Φ 6 × 0.3 mm, and the thickness–diameter ratio was 0.05. The heat pipe was manufactured via a complex procedure. The initial pipe was prepared by multiple passes of the float-plug drawing (FPD) process followed by an annealing treatment. To obtain a semi-hardening state, one more FPD process was imposed on the pipe with a 30–35% area reduction. The heat pipes billet was obtained with a size of Φ 6 × 0.3 mm. Next, 980 °C × 3.5 h sintering treatment was carried out in a vacuum furnace after copper powder was filled into the billet. Finally, the pipes were filled with special liquids and sealed to make an internal vacuum, and then the heat pipes were finished. To simulate the rotary bending performance in this experiment, a bending radius of R = 10.5 mm was chosen, and the bending angle was 180 degrees.

The microstructure of the pipe wall after bending deformation was observed with an optical microscope (OM) at view sections A and B; the locations from where the samples were taken can be seen in Figure 1. The scanning electron microscopy (SEM) analysis was carried out with an FEI Nova Nano scanning electron microscope (FEI Ltd., Hillsboro, OR, USA) to observe the whole macrostructure of the orange peel morphology. The observation area was selected at the maximum bent part, as marked by the black dotted box in Figure 1. The samples were cleaned with ultrasonic waves in an alcohol solution and then directly observed without the grinding process. This method allowed the original state of the surface roughness to be preserved. To investigate the local 3D profile of the bulge of the bent surface, a micro-XAM surface mapping microscope was used to measure the surface properties. Because the whole bent area was too large to measure, and the bent surface led to different distances to the lens, so the local section of the bent area was chosen for observation. The scan area was a square with a side length of 728 μm, and the scan step was 1.45 nm. The crystallographic orientations of the orange peel section were measured with electron backscatter diffraction (EBSD) on an FEI Nova Nano SEM 430 field-emission scanning electron microscope (FEI Ltd., Hillsboro, OR, USA) equipped with fully automatic HKL technology. The EBSD samples for EBSD measurements were cut from a bending surface (shown in Figure 1) 5 mm in length (DD) and 3 mm in width (CD), and from the surface of sintered pipes 5 mm in length (DD) and 2 mm in width (CD). The samples were slightly ground and then polished, followed by an electro-polishing treatment. The local coordinate system of the EBSD sample was defined as shown in Figure 1, and the DD–CD section was set as the observation plane. The EBSD maps were acquired using a step size of 6 µm to analyze the grains and annealing twins. The indexing rate was 100%. In this study, a critical misorientation angle of 2 °C was applied to observe boundaries in the orientation maps, where low-angle grain boundaries (LAGBs) and high-angle grain boundaries (HAGBs) were defined as boundaries between grains with misorientation 2–10 and >10 °C, respectively.

## 3. Results and Discussion

### 3.1. Microstructure of the Heat Pipe after Bending Deformation

The heat pipe underwent high-temperature sintering treatment at 980 °C × 3.5 h, resulting in full recrystallization and severe grain growth. The microstructure of the heat pipe at the bent area along the longitudinal and transverse slices, marked as sections A and B (Figure 1), respectively, were examined using an optical microscope (OM), as shown in Figure 2. To provide a comprehensive view of the pipe microstructure, multiple images were combined and stitched together. The band-shape annealing twins present in the matrix grains could be clearly observed. FCC metals of low-to-medium stacking fault energy tend to have a preponderance to form twin boundaries in the recrystallization process [16,17,18]. Pure copper with medium stacking faults also prefers to form annealing twins during recrystallization, and the details were reported by Field [19]. Annealing twins show the difference in brightness after etching due to their unique orientation relationship with the matrix, as shown in Figure 2a,b, marked in a white dotted circle. The distinct grain boundaries are highlighted in the green circles in Figure 2a, showing that only one layer of grain was present in the pipe wall. The porous structure formed by copper powder sintering can be seen on the inner side of the pipe wall, which provides certain support to the inner side of the bending and prevents wrinkling. It can be concluded that the grains severely coarsened during the sintering treatment. These annealing twins grew with the matrix grains and became broader in width and reach the whole wall thickness in length. At the bent section, the annealing twins bulged out of the outer side surface of the heat pipe. Figure 2b shows an enlarged schematic diagram of the bulged annealing twin; ∆d is the value of the distance of the bulge. The protruding height ∆d of the bent area to the surface was approximately 10–20 µm, which contributed to the “orange peel” morphology.

### 3.2. Characterization of Orange Peel Morphology on the Bent Surface

Figure 3 shows the outer surface at the maximum bending position of the orange peel pipe and the normal pipe. It was easy to see the orange peel defect by visual inspection as it presented an uneven surface and a dark look (Figure 3a), while the normal pipe had a smooth surface (Figure 3c). An uneven surface strongly affects the quality of the heat pipe product. A backscatter electron image (BSE) was used to examine the micro-surface topography of the bent zone marked by the black box in Figure 3a,c. The bent section observed at a magnification of 50× showed an uneven surface result from the band-like bulge, as marked by white arrows. The bent section on the normal pipe showed slip bands (marked by yellow arrows) produced during the bending deformation, which was about 45 degrees to the longitudinal direction. The remaining slip band on the surface was an indication of the good plasticity of the normal pipe, and the direction of the slip bands revealed the mainly tensile stress along the longitudinal direction. The results confirm the conclusion obtained by Romanova [4]: the smallest out-of-plane surface displacements are attributed to intra-grain dislocation glide, and the larger displacements are associated with the movement of small grains and give rise to the formation of the orange peel pattern. In the copper pipes, after the sintering treatment, the annealing twins grew with the recrystallized grains. During the bending process, the outer side surface of the pipe was mainly subjected to tensile stress. These annealing twins could not coordinate with the matrix grains during deformation, interrupted the slipping of dislocation, and hindered plastic deformation.

In order to investigate the 3D profile of the bulge on the surface, a surface mapping microscope was used to examine section A (Figure 3b), with a size of 728 × 728 μm. Figure 4a shows the “hill and valley” morphology that was due to the fluctuation at various heights (Z-axis direction), which is indicated by the variable color legend in Figure 4c. There was a gradual decrease in height at the edge because of the curved surface, but it did not affect the observation of the bulge area. It could be observed that the band-like annealing twins protruded from a surface with clear grain boundaries, which corresponds to the results measured in Figure 2b. Figure 4c shows the height undulations along with the dotted line AB in the middle of the region. The peak values of the curve in Figure 4c appear at the positions corresponding to the annealing twins. The bulged twins are mostly perpendicular to the DD, being about 200–300 μm in width and about 10 μm in height. The micro-characteristic feature of valleys and hills formed the orange peel defects on the bent surface of the heat pipe.

Electron backscatter diffraction (EBSD) observations were carried out to further examine the grain orientations of the bent surface. A sample was taken from the bent section, as shown in Figure 3a, and the DD–CD plane was taken as the observation plane for EBSD measurement. Figure 5a illustrates the grain orientations of the orange peel surface; the annealing twins in purple (marked by black arrows) show an obvious difference in orientation from the matrix grains in green. These specially oriented “band-like” annealing twins exhibit a sharply discordant orientation with the matrix grains. Figure 5b shows the distribution of misorientation angle that the curve presents as two peaks, which correspond to the low angle grain boundaries (LABs) and the twin-relationship (Σ3) boundaries, respectively. During the grains coarsening process, the matrix grains encountered each other and formed LABs, as seen in Figure 5a. The texture components were analyzed by pole figure based on the EBSD information, as shown in Figure 5c [20]. The observation plane was defined by the DD–CD surface (the normal direction is RD) with the EBSD measurement. As a result, we could determine the relationship between the sample coordinate system and the crystal coordinate system as follows: for example, the Goss texture component means that the {110} crystal plane parallel to the observation plane (DD-CD plane), and the <001> crystal direction parallel to the DD direction.

### 3.3. Deformation Inhomogeneity between Annealing Twins and Matrix Grains

The above results indicate that these bulge regions on the surface originated from the annealing twins. Therefore, a sample of peel pipe before bending deformation was prepared for EBSD characterization, as shown in Figure 6a. The observation plane was also on the DD–CD surface. The black lines represent high-angle grain boundaries (with a misorientation angle > 10°). From the IPF map, it is obvious that the matrix grains in green severely grew and finally combined so that the low-angle grain boundaries were removed. In addition, the texture components were similar to the bent tube; that is, the Goss component of green grains was the main texture, accompanied by an amount of copper texture composed of the annealing twins. Compared with the bent tube, the color of the twins with copper texture in the pre-bent tube was somewhat different, which was due to the orientation deviations. In this study, the allowable deviation range for texture calculation was within 10 °C from the ideal texture orientation. The orientations of the crystal may have changed a bit due to the bending process, but it could not change the texture type with such a small deformation. It was confirmed that the matrix and the annealing twins formed the Goss and copper textures, respectively, as a result of the sintering treatment. Thus, the annealing twins in purple/pink contributed to the discordant orientation with the matrix grains, i.e., the “special orientation”. It was noticed that another annealing twin variant in green orientation existed in the matrix grains, which disappeared after bending deformation. The 3D crystal cubic of orientation components is illustrated in Figure 6a. The relationships between the two types of annealing twins (defined as T1 and T2) were calculated at about 59.9 <111> and 55.9 °C <111>, respectively. The annealing twin variants were nucleated on the {111} crystal plane as a growing accident because of the low stacking faults.

It is known that slip is the main deformation type of pure copper with five independent {111} <110> slip systems. With the grain-orientation information obtained from EBSD, it is possible to measure the activity of the slip system by the Schmid factor (SF). In this study, the SF calculation was important for understanding the intergranular-scale deformation behavior. It is known that the SF value is dependent on the direction of the applied stress and the deformation mode of certain grain orientations. The loading condition of the pipe during the rotary bending process is complex because of the stress and strain gradients that exist in the pipe. The analysis of stress distribution has been researched by both theoretical calculation and by finite element simulation methods [21,22,23]. However, it is difficult to measure the accurate stress state during the bending process because macroscopic stress is not constant. Moreover, the local stresses among individual grains are also various. From the previous results, the stress tensor can be reduced to a set of principal stresses: major stress along DD (σ_1_), stress along CD (σ_2_), and the stress along the normal direction (σ_3_) of the observation plane (DD–CD). σ_1_ is assumed to be the main stress leading to the tensile strain of the extrados surface, i.e., σ_1_ >> σ_2_ and σ_3_, which is equivalent to uniaxial stress for SF value calculations. The SF value does not have strict physical meaning, but it was helpful to evaluate, on a comparative basis, the deformation resistance as well as the deformation uniformity in this study [24].

Figure 6b shows the SF distribution of the peel pipe before bending deformation. The value of SF was equal in the individual matrix grain, except for the part divided by the annealing twins. From the result of the Schmidt factor calculation, there were three peaks in the curve. Compared with Figure 6a, the highest peak corresponds to the matrix grains, with the maximum Schmidt factor value of about 0.43. The next two small peaks with values of about 0.36 and 0.28 correspond to the pink and purple annealing twins and part of the green annealing twins, respectively. The SF value of the annealing twins was lower than the matrix grains on average. The SF fraction curve with several peaks shows the inhomogeneous deformation behavior among the grains. The SF is used to evaluate the activation of deformation modes, i.e., with the highest SF, the slip system has the greatest possibility of being activated. The existence of annealing twins in the matrix decreases the uniformity of the distribution of SF; that is, it decreases the uniformity of the deformation of grains. Moreover, the slip propagation across the grain boundary is primarily connected with the neighboring grains’ orientation (the type of grain boundary) and their relation with respect to the direction of operating forces. It was confirmed that dislocation transmission through twin-type boundaries is never a direct transfer [25]. Dislocation pile-ups or decomposition in the boundary occurs, which makes the twins act as an effective barrier to slip.

From above, it was concluded that the inhomogeneity of the deformation was created by the presence of annealing twins from the pre-bent condition, and then they further influenced the deformation during bending. After the sintering process, the matrix grains seriously coarsened and merged together. The annealing twin with the unique twin-type boundary relationship may have hindered the slipping of dislocation, which could lead to the deformation inhomogeneously in grains, as seen in Figure 7. The inhomogeneous intergranular deformation caused the uneven surface of the bent tube, the so-called orange peel morphology.

## 4. Conclusions

In this work, the micro- and macro-structure and forming mechanism of the orange peel defect on the bent surface of a heat pipe were studied. The main conclusions can be summarized as follows:(1)The orange peel defect of heat pipes is caused by inhomogeneous intergranular deformation between the annealing twins and matrix grains, which exhibit nonuniform size and crystal orientation. The annealing twins exhibit a hard orientation in bending deformation and may bulge out on the pipe surface, which presents an uneven morphology.(2)The matrix grains seriously coarsen and form a strong {110} <001> Goss texture during high-temperature sintering treatment, while the annealing twins in it show a {112} <111> copper texture. Thus, the annealing twins act as a barrier for the slip of dislocations by both the unique orientation with the matrix grains and the twin-type boundaries.(3)According to the above results, it is suggested that orange peel defects on bending surfaces of heat pipes can be relieved by reducing or eliminating the proportion of “hard-oriented” annealing twins or by inhibiting recrystallized grains’ growth, such as by adding alloying elements.

## Figures and Tables

**Figure 1 materials-16-02147-f001:**
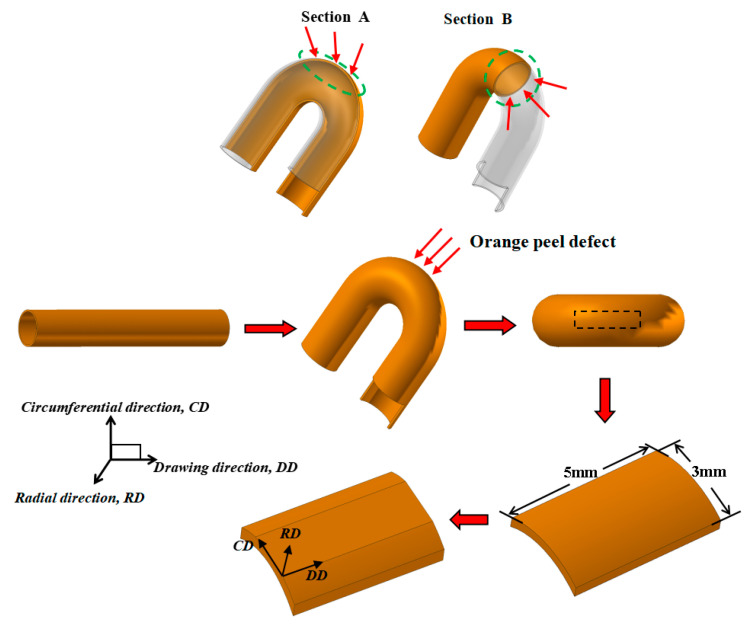
The selection and location of samples on the bent heat pipe.

**Figure 2 materials-16-02147-f002:**
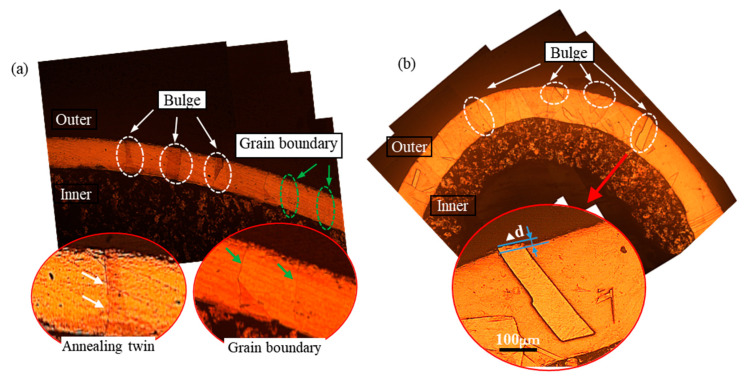
The microstructure of the outside of the bent pipe along the longitudinal slice: (**a**) section A and transverse slice; (**b**) section B and the enlarged schematic diagram of the pipe wall bulge.

**Figure 3 materials-16-02147-f003:**
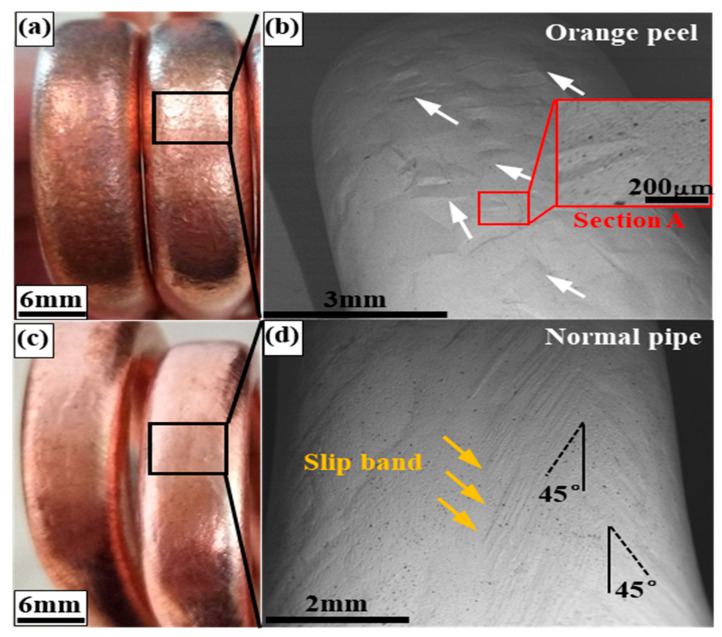
Orange peel morphology on the bending section of the heat pipe (**a**,**b**) and the compared no-peel pipe (**c**,**d**) by visual inspection (**a**,**c**) and by SEM observation (**b**,**d**).

**Figure 4 materials-16-02147-f004:**
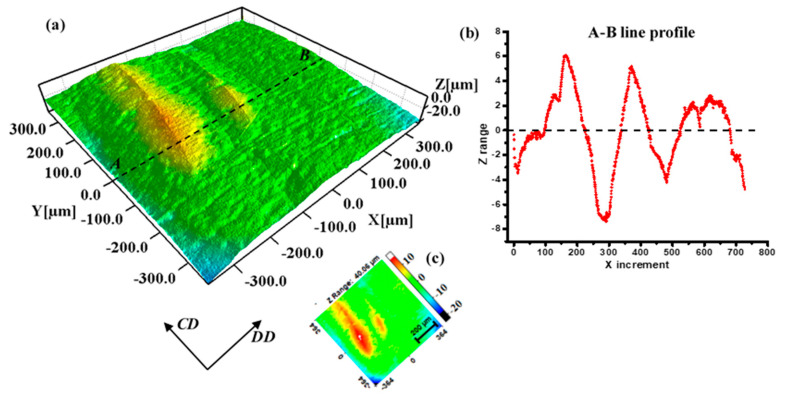
The 3D profile of the bulged grains on the orange peel surface: (**a**) false color topography image; (**b**) distribution of Z range along line AB; (**c**) legend of the 3D profile.

**Figure 5 materials-16-02147-f005:**
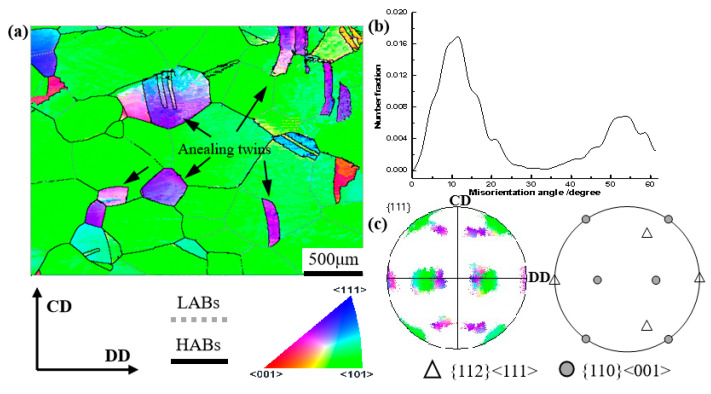
EBSD maps, pole figure, and misorientation distribution of the bent surface of the peel pipe: (**a**) IPF; (**b**) misorientation angle distribution; (**c**) pole figure.

**Figure 6 materials-16-02147-f006:**
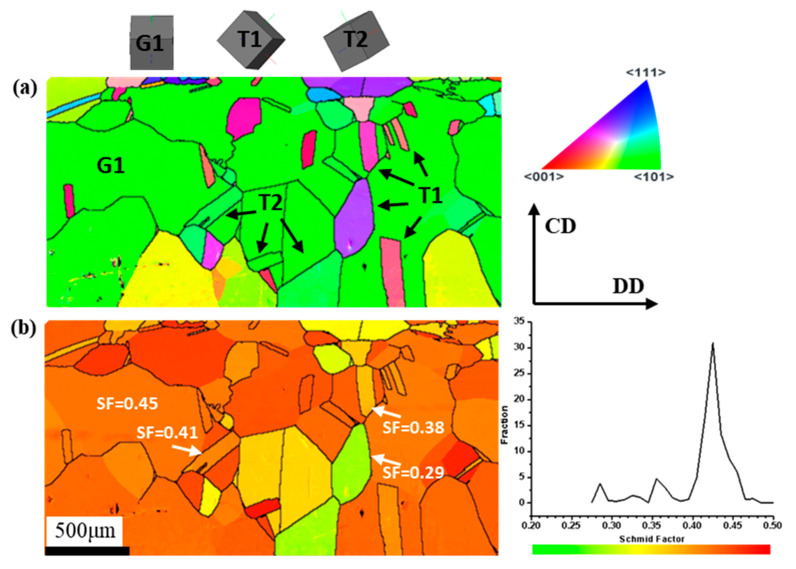
EBSD maps and Schmid factor distribution of peel pipe before bending deformation: (**a**) IPF; (**b**) Schmid factor distribution.

**Figure 7 materials-16-02147-f007:**
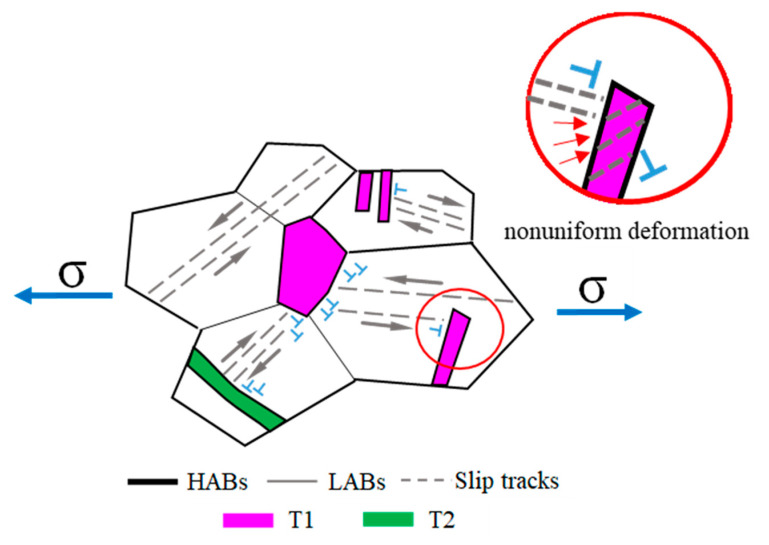
Schematic diagram of the intergranular nonuniform deformation subjected to the uniaxial stress state.

## Data Availability

Not applicable.

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
