# Peer review of "Special-Oriented Annealing-Twins-Induced Orange Peel Morphology of Heat Pipe under Bending Deformation"

_materials, 2023, doi:10.3390/ma16062147_

Round 1

Reviewer 1 Report

The last sentence of the first paragraph of the Introduction is key to the potential value of this work. The authors give no references or reasoning why either formability or decreased heat dissipation could cause any serious failures. It is just an opinion unsupported by references or facts. For the formability issue, there is no evidence given that the formability is affected to the point where parts are made and suffer mechanical failures such as cracking or fracture. The surface morphology might not be pretty, but that doesn’t ultimately matter. For the case of a decrease in heat transfer efficiency, the authors have not supported their opinion with any logic or references. A small amount of surface irregularity probably has almost zero effect on heat transfer efficiency because, (1) the fraction of surface area with this morphology is likely to be small enough to be insignificant (the authors did not explore this), and (2) surface irregularities could act like small pins or fins which are know to be used on purpose to increase efficiency of heat exchangers by increasing the surface area for heat transfer. The authors must show references or calculations or finite element results to support their claim of decreased heat dissipation efficiency. The authors repeat this in the last paragraph of the Introduction, stating that this “morphology on the outer surface becomes a main problem”. No problem has been shown, it is purely the opinion of the authors and not supported by references of facts. Please provide references that state that the orange peel morphology results in insufficient mechanical properties to produce certain required geometries of bends and/or that heat transfer is less efficient to the point that sintered pure copper is not sufficient for certain heat transfer applications.

In the second paragraph of Section 2 that authors state that Figure 1 shows optical microscope images. Figure 1 shows a series of drawings of where samples were taken from the 180 degree bend specimen.

In the first paragraph of Section 3, the authors begin a sentence, “As is known to all”. Please remove this, as it is not only completely inaccurate, it is also completely superfluous. If it is known to all, then why put it in?

Figure 2a needs better contrast. It is of insufficient quality and resolution to support your points relative to it.

In Sub-Section 3.2, first paragraph, sentence 3, the authors once again claim, without references or facts, that heat conduction will decrease. Please explain because what you provide is just an opinion. If this tube is exchanging heat between two fluids, for instance, a small amount of surface roughness on a very small area of tubing would have a negligible effect on heat transfer. The authors mention adhesion, but surface roughness frequently enhances adhesion of things such as epoxies and solders. Surface cleanliness is generally much more important for adhesion, unless these tubes are being cold-welded to other metals, but that is not mentioned by the authors.

At the end of page 5, penultimate sentence, the authors claim that the bulged twins are from 10-20 micrometers high. There is no resolvable evidence that they could be 20 micrometers high. I can resolve 10 on Figure 4(c), but certainly not 20.

Author Response

Point 1: The last sentence of the first paragraph of the Introduction is key to the potential value of this work. The authors give no references or reasoning why either formability or decreased heat dissipation could cause any serious failures. It is just an opinion unsupported by references or facts. For the formability issue, there is no evidence given that the formability is affected to the point where parts are made and suffer mechanical failures such as cracking or fracture. The surface morphology might not be pretty, but that doesn’t ultimately matter. For the case of a decrease in heat transfer efficiency, the authors have not supported their opinion with any logic or references. A small amount of surface irregularity probably has almost zero effect on heat transfer efficiency because, (1) the fraction of surface area with this morphology is likely to be small enough to be insignificant (the authors did not explore this), and (2) surface irregularities could act like small pins or fins which are know to be used on purpose to increase efficiency of heat exchangers by increasing the surface area for heat transfer. The authors must show references or calculations or finite element results to support their claim of decreased heat dissipation efficiency. The authors repeat this in the last paragraph of the Introduction, stating that this “morphology on the outer surface becomes a main problem”. No problem has been shown, it is purely the opinion of the authors and not supported by references of facts. Please provide references that state that the orange peel morphology results in insufficient mechanical properties to produce certain required geometries of bends and/or that heat transfer is less efficient to the point that sintered pure copper is not sufficient for certain heat transfer applications.

Response 1: Thanks for the reviewer’s comments, and the reviewer is right about the shortage of relevant references of the serious failures caused by orange peel morphology.

The authors would explain that the topic of this study is based on a practical engineering problem proposed by an enterprise. The surface orange peel problem has become the key factor affecting the product qualification rate, and is also the main reason why customers refuse to accept the product. The authors learned from the investigation that the harmfulness of surface roughening is mainly manifested in the following aspects: (1) Affecting the beauty of the product, which makes it difficult to be used in the field of high-end equipment heat dissipation; (2) The heat exchange is affected. When the heat pipe is used, one end is usually connected with a heat source such as CPU, and the other end is connected with a cold source such as a fan, as seen in Fig. 1. The uneven surface of the bending part will affect the fitting degree of the outer surface of the heat pipe and the connecting surface, which is easy to produce voids, thus affecting the heat transfer; (3) The influence on the forming ability is mainly reflected in the decline of surface quality after bending deformation.

Nevertheless, the authors agree with the suggestions put forward by the reviewer, and actually the relevant references are few, the authors has revised the manuscript accordingly.

Point 2: In the second paragraph of Section 2 that authors state that Figure 1 shows optical microscope images. Figure 1 shows a series of drawings of where samples were taken from the 180 degree bend specimen.

Response 2: Thanks for the reviewer’s comment. The reviewer is totally right and the manuscript has been revised.

Point 3: In the first paragraph of Section 3, the authors begin a sentence, “As is known to all”. Please remove this, as it is not only completely inaccurate, it is also completely superfluous. If it is known to all, then why put it in?

Response 3: Thanks for the reviewer’s comment. The manuscript has been revised.

Point 4: Figure 2a needs better contrast. It is of insufficient quality and resolution to support your points relative to it.

Response 4: Thanks for the reviewer’s comment. The Fig. 2 has been redraw and an enlarged view has been added.

Point 5: In Sub-Section 3.2, first paragraph, sentence 3, the authors once again claim, without references or facts, that heat conduction will decrease. Please explain because what you provide is just an opinion. If this tube is exchanging heat between two fluids, for instance, a small amount of surface roughness on a very small area of tubing would have a negligible effect on heat transfer. The authors mention adhesion, but surface roughness frequently enhances adhesion of things such as epoxies and solders. Surface cleanliness is generally much more important for adhesion, unless these tubes are being cold-welded to other metals, but that is not mentioned by the authors.

Response 5: Thanks for the reviewer’s comment. The manuscript has been revised accordingly.

Point 6: At the end of page 5, penultimate sentence, the authors claim that the bulged twins are from 10-20 micrometers high. There is no resolvable evidence that they could be 20 micrometers high. I can resolve 10 on Figure 4(c), but certainly not 20.

Response 6: Thanks for the reviewer’s comment. The authors checked the coordinate axis value and revised the sentence.

Reviewer 2 Report

Dear Authors,

1. What recommendations can be made based on the results of this article in order to reduce the rejection of heat pipes due to orange peel in production? This answer can be added as a conclusion to the article.

2. Many links to articles older than 5 years. Please add earlier articles to the list of references.

Best regards!

Author Response

Point 1: What recommendations can be made based on the results of this article in order to reduce the rejection of heat pipes due to orange peel in production? This answer can be added as a conclusion to the article.

 Response 1: Thanks for the reviewer’s suggestion. The manuscript has been modified as follow:

According to the above results, it is suggested that the orange peel defect on the bending surface of heat pipe can be relieved by reducing or eliminating the proportion of “hard oriented” annealing twins, or by inhibiting the recrystallized grains growth, such as adding alloying elements.

Point 2: Many links to articles older than 5 years. Please add earlier articles to the list of references.

Response 2: Thanks for the reviewer’s comments, the earlier references has been added.

  1. Nie, N.; Su, L.H.; Deng, G.Y.; Li, H.J.; Yu, H.L.; Tieu, A.K. A review on plastic deformation

induced surface/interface roughening of sheet metallic materials. J. Mater. Res. Tec., 2021, 15,

6574-6607.

  1. Sunal, A.P.; Yasin, K.K.; Onur, K. On the utilization of Sachs model in modeling deformation of

surface grains for micro/meso scale deformation processes. J. Manuf. Process, 2021, 68, 1086-1099.

  1. Grzegorz, S.; Ryszard, B.; Bartlomiej Z. Deformation‑induced roughening by contact

compression in the presence of oils with diferent viscosity: experiment and numerical simulation.

Tribol. Lett, 2020, 68, 117.

  1. Cai, Y.; Wang, X.S.; Du, Y. Surface roughening behavior of the 6063-T4 aluminum alloy during

quasi-in situ uniaxial stretching. Materials, 2022, 15, 6265.

Reviewer 3 Report

In the introduction you wrote not only about the importance of explaining the occurence of a surface defect, but also "to provide a reference for production". However, there is not a word in the conclusion about how the authors propose to avoid the defect in question.

Author Response

Point 1: In the introduction you wrote not only about the importance of explaining the occurence of a surface defect, but also "to provide a reference for production". However, there is not a word in the conclusion about how the authors propose to avoid the defect in question.

Response 1: Thanks for the reviewer’s suggestion. The manuscript has been modified as follow:

According to the above results, it is suggested that the orange peel defect on the bending surface of heat pipe can be relieved by reducing or eliminating the proportion of “hard oriented” annealing twins, or by inhibiting the recrystallized grains growth, such as adding alloying elements.

Reviewer 4 Report

The authors have discussed the orange-peel morphology of the bent pipes with the help of orientation of annealing twins and SF. As calculating SF for the bent pipes are difficult, so SF of a pre-bent pipe is used to explain the inter granular and inhomogeneous deformation of the grains, that result in uneven surface. The topic is of interest, experimentation is adequate, but few explanations are missing. 

Also, authors need to check thoroughly the text as there are several grammatical mistakes, but few of them are pointed out here.  

1. Line 76: 'grinding' process, not 'grind'.

2. Section 2: Can authors include a bit of details about the EBSD parameters (i.e. step size, indexing rate, misorientation angles for HAGB and LAGB etc.) used?

3. Fig 1: annotation should be 'radial' direction, not 'radical'.

4. Fig 2b: include the scale bar in the bottom image.

5. Grammatical mistakes, such as (i) 'sintering process', not sinter process, (ii) 'mainly subjected to', not main subjected to' etc. Please check them thoroughly in the manuscript.

6. Fig 3b: include the scale bar on section A image, and a bigger image would be better to visualize the peel section

7. Authors should include the texture analysis of the pre-bent pipe to conclude that the matrix and the annealing twins form the Goss and copper textures respectively as a result of sintering. This will improve the quality of the work. 

8. Fig 5a: The IPF shows twins in purple and the matrix in green. Can the authors explain a bit about the few twins in blue-coloured grains? Are they from before the bending condition?

9. Line 184: Do the authors mean that the matrix grains being coarser before bending? It is difficult to compare the grains from the IPFs before and after bending because of the different scales. Can authors use same scales for Figures 5a and 6a?

10. Line 187: there are no annealing twins present in the purple coloured grain in Fig 6a. It is not clear if the authors meant to say Fig 5a or 6a.

11. Line 215: 'inhomogeneous deformation', not 'inhomogeneous of deformation.

12. Line 225: 'seriously coarsen', not 'serious, re-structure the sentence.

13. Can authors include the area fraction maps for the recrystallized, sub-structured, and deformed grains (from EBSD analysis) for pre-bent and bent pipe materials? That will support the current findings that the authors have presented. 

14. In the discussion, it is not clear if the inhomogeneity of the deformation is created by the presence of twins from the pre-bent condition and then they further influence the deformation during bending. Can authors comment on that?

Author Response

Point 1: The authors have discussed the orange-peel morphology of the bent pipes with the help of orientation of annealing twins and SF. As calculating SF for the bent pipes are difficult, so SF of a pre-bent pipe is used to explain the inter granular and inhomogeneous deformation of the grains, that result in uneven surface. The topic is of interest, experimentation is adequate, but few explanations are missing.

Response 1: Thanks for the reviewer’s comments. The manuscript has been supplemented accordingly.

From the result of Schmidt factor calculation, there are three peaks in the curve. Compared with Fig. 6a, the highest peak corresponds to the matrix grains , with the maximum Schmidt factor value is about 0.43. The next two small peaks with the value about 0.36 and 0.28 are corresponding to the pink color annealing twins and the purple color and part of green color annealing twins, respectively. The SF value of the annealing twins is lower than the matrix grains in average. The SF fraction curve with several peaks show the inhomogeneous deformation behavior among the grains.

Point 2: Line 76: 'grinding' process, not 'grind'.

Response 2: Thanks for the reviewer’s comments. The manuscript has been revised.

Point 3: Section 2: Can authors include a bit of details about the EBSD parameters (i.e. step size, indexing rate, misorientation angles for HAGB and LAGB etc.) used?

Response 3: Thanks for the reviewer’s comments. The EBSD parameters has been added as follows:

The EBSD maps were acquired using step size of 6µm to analyze the grains and annealing twins. The indexing rate was 100%. In this study, a critical misorientation angle of 2°was applied to observe boundaries in the orientation maps, where low-angle grain boundaries (LAGBs) and highangle grain boundaries (HAGBs) were defined as boundaries between grains with misorientation 2-10°and >10°, respectively.

Point 4: Fig 1: annotation should be 'radial' direction, not 'radical'.

Response 4: Thanks for the reviewer’s comments. The Fig. 1 has been revised.

Point 5: Fig 2b: include the scale bar in the bottom image.

Response 5: Thanks for the reviewer’s comments. The Fig. 2 has been revised.

Point 6: Grammatical mistakes, such as (i) 'sintering process', not sinter process, (ii) 'mainly subjected to', not main subjected to' etc. Please check them thoroughly in the manuscript.

Response 6: Thanks for the reviewer’s comments. The sentences mentioned has been revised and the text has been checked.

Point 7: Fig 3b: include the scale bar on section A image, and a bigger image would be better to visualize the peel section

Response 7: Thanks for the reviewer’s comments. The Fig. 3 has been revised.

Point 8: Authors should include the texture analysis of the pre-bent pipe to conclude that the matrix and the annealing twins form the Goss and copper textures respectively as a result of sintering. This will improve the quality of the work.

Response 8: Thanks for the reviewer’s comments. The manuscript has been revised as follow:

In addition, the texture components are similar to the bent tube, that is, the Goss component of green grains is the main texture, accompanied by a amount of Copper texture composed of the annealing twins. Compared with the bent tube, the color of the twins of Copper texture in the pre-bent tube is somewhat different, which is due to the orientation deviations. In this paper, the allowable deviation range for texture calculation is within 10 ° from the ideal texture orientation. The orientations of the crystal may change a bit due to the bent process, but can not change the texture type with such a small deformation. It is confirmed that the matrix and the annealing twins form the Goss and Copper textures respectively as a result of sintering treatment.

Point 9: Fig 5a: The IPF shows twins in purple and the matrix in green. Can the authors explain a bit about the few twins in blue-coloured grains? Are they from before the bending condition?

Response 9: Thanks for the reviewer’s comments. 

In this paper, the allowable deviation range for texture calculation is within 10 °from the ideal texture orientation. As a result, the deviation of this part of twins may lead to the difference in colors. Also, the deviation may be caused by the tube flattening or grinding during the EBSD sample preparation. These twins are from before the bending condition. The authors think it is doesn’t matter for the judgment of the texture.    

Point 10: Line 184: Do the authors mean that the matrix grains being coarser before bending? It is difficult to compare the grains from the IPFs before and after bending because of the different scales. Can authors use same scales for Figures 5a and 6a?

Response 10: Thanks for the reviewer’s comments. The scales of Fig. 5a and Fig. 6a has been redraw. Actually, the grain size of the pipes before and after the bending process are the same scales.  

Point 11: Line 187: there are no annealing twins present in the purple coloured grain in Fig 6a. It is not clear if the authors meant to say Fig 5a or 6a.

Response 11: Thanks for the reviewer’s comments. The twins mentioned in Fig. 6a should be pink colored. The manuscript has been revised.

Point 12: Line 215: 'inhomogeneous deformation', not 'inhomogeneous of deformation.

Response 12: Thanks for the reviewer’s comments. The manuscript has been revised.

Point 13: Line 225: 'seriously coarsen', not 'serious, re-structure the sentence.

Response 13: Thanks for the reviewer’s comments. The manuscript has been revised.

Point 14: Can authors include the area fraction maps for the recrystallized, sub-structured, and deformed grains (from EBSD analysis) for pre-bent and bent pipe materials? That will support the current findings that the authors have presented.

Response 14: Thanks for the reviewer’s suggestion. The authors tried to provide the area fraction maps for the recrystallized, sub-structured, and deformed grains, but it is hard to obtain a good picture. Because the tube wall sample is very thin and the observation area is large, some scratches are inevitably left during the grind and polish process, which affects the statistics of recrystallized, sub-structured, and deformed fraction. Therefor, it is can be seen that the pipe was fully recrystallized after sintering process, and after bending deformation the fraction may change a little.  

Point 15: In the discussion, it is not clear if the inhomogeneity of the deformation is created by the presence of twins from the pre-bent condition and then they further influence the deformation during bending. Can authors comment on that?

Response 15: Thanks for the reviewer’s comments and suggestions. The manuscript has been revised.

Round 2

Reviewer 1 Report

Thank you for expanding the work, it is potentially very useful as this type of bending for heat exchangers is common in industry and problems do arise from this issue

Author Response

Thanks for the reviewer's comment.

Reviewer 4 Report

The authors have addressed the comments and incorporated the changes. Still, some grammatical mistakes are there, which need to be corrected. 

Author Response

Thanks for the reviewer’s comments. The grammatical mistakes in manuscript has been corrected. Please see the attachment.
